# Mobility Control Centre and Artificial Intelligence for Sustainable Urban Districts

Francis Marco Maria Cirianni [1,*], Antonio Comi [2] and Agata Quattrone [1]

1   Department of Civil, Environmental and Mechanical Engineering, University Mediterranea,
    89100 Reggio Calabria, Italy; agata.quattrone@gmail.com
2   Department of Enterprise Engineering, University of Rome Tor Vergata, 00133 Rome, Italy
*   Correspondence: francis.cirianni@unirc.it

**Abstract:** The application of artificial intelligence (AI) to dynamic mobility management can support the achievement of efficiency and sustainability goals. AI can help to model alternative mobility system scenarios in real time (by processing big data from heterogeneous sources in a very short time) and to identify network and service configurations by comparing phenomena in similar contexts, as well as support the implementation of measures for managing demand that achieve sustainable goals. In this paper, an in-depth analysis of scenarios, with an IT (Information Technology) framework based on emerging technologies and AI to support sustainable and cooperative digital mobility, is provided. Therefore, the definition of the functional architecture of an AI-based mobility control centre is defined, and the process that has been implemented in a medium-large city is presented.

**Keywords:** artificial intelligence; smart mobility; transport; sustainable districts; intelligent transport systems; mobility as a service; mobility control centre; urban mobility

## 1. Introduction

In the past, technologies have helped organizations to tackle problems in a variety of fields of business, including retail, finance, insurance, healthcare, and even in sports. Some of these have altered the way companies are run by lowering operating costs and increasing efficiency. One of the industries where technology has always made a difference and led to evolution is transportation, and the latest cutting-edge technology is being effectively used is transportation to overcome difficulties, such as traffic congestion, unforeseen delays, and routing, which affect not only the efficiency of the vehicle but also the effectiveness of the system.

In fact, mobility is undergoing a profound change ensured by digital innovation [1]. Emerging technologies applied to the transport sector allow for personalized and dynamic management of mobility and guarantee a new balance between efficiency and sustainability in our cities [2].

Artificial intelligence (AI) can represent one of the fundamental levers for decision making and for a dynamic and continuous readjustment of transport supply for an increasingly flexible and fluid mobility demand, through a new generation of digital platforms and mobility urban centres in the era of Cooperative–Intelligent Transportation Systems (ITSs). Furthermore, AI combined with the acquisition of big data and traffic flows from field systems and video analysis systems (both at the edge and central levels) can be usefully used to calibrate demand simulation models and interpret phenomena, not only in ordinary conditions but also during exceptional/disruption events [3–5].

The majority of the world's large cities struggle with traffic and logistics-related difficulties. This is brought on by a series of factors, including the increasing number of vehicles per capita on the road and the increasing demand in people and freight trips on the network. Technology might be of great help in designing and operating a sustainable transportation system [6]. Urban regions struggle with traffic congestion, but artificial

intelligence (*AI*) technologies can contribute to more effective traffic management to acquire real-time information from moving cars and employ mobility on demand for trip planning through a single user interface. Other options for effective traffic management include the secure integration of AI-based decision making, routing, transportation network services, and other mobility optimisation technologies.

AI applications in the field of transport systems have so far mainly been used in the field of autonomous vehicles and in the management and maintenance of physical infrastructure. However, AI can also be applied very effectively in both the planning and operational phases, for the implementation of demand management policies and supply adaptive and intermodal transport services, according to the MaaS (Mobility as a Service) approach [7].

New AI solutions for the analysis of information enable effective decision support, which, combined with the logic of process automation and remote ITS management, allows for a radical change in the dynamics of mobility management [8,9]. This produces, on the one hand, a reduction in costs and errors and, on the other hand, the customisation of door-to-door mobility services [10].

AI can help in the real-time diagnosis of the mobility system by processing big data from heterogeneous sources in a very short time and by comparing phenomena in similar contexts, identifying network and service configurations, and supporting decision making in implementing measures for the management of demand that allows one to achieve the objectives of a public decision maker in the collective interest—*system optimum* (i.e., reduction in community costs and externalities, minimization of the level of congestion, maximization of user satisfaction, improvement in safety, reduction in gas emissions and greenhouse and atmospheric pollutants).

In this context, AI-based solutions can push towards a significant advancement to city mobility in all cities within all countries. Some basic features, such as collecting data, storing data, and analysing data, can be integrated in order to improve the existing urban transport systems. This paper proposes a contribution in answer to the following open questions:

- How can AI contribute to the promotion of sustainable urban mobility by improving the users' travel experience and shifting towards more sustainable means of transport (Section 2)?
- How can urban sustainable mobility planning benefit from the opportunity offered by AI to support decision making for identifying and implementing sustainable measures (Section 3)?
- What are the main features of a mobility control centre in the promotion of sustainable mobility (Sections 4 and 5)?

Therefore, starting from an in-depth and model-based analysis of the application cases for the different automation techniques, this paper reviews the current state of the practice in AI and telematics in supporting improvements in the sustainability and liveability of urban and metropolitan areas, explores the capacity of AI in building smart digital solutions, also elaborating some key recommendations for city scientists, policymakers, transport, and urban planners, and an agenda for future research directions. In addition, it presents how such findings are guiding the development of an AI-based Mobility Control Centre in a medium-large city. This analysis, in particular, provides innovative AI techniques for each application layer and macro-functionality of a Mobility Control Centre, specifying their applicability starting from the existing literature.

The results show how AI is bringing the urban mobility sector closer through Mobility-as-a-Service (MaaS) schemes and illustrate, with a virtual district, how such AI-enabled technological advances linked with public policies can help design sustainable urban mobility to provide a better travel experience for city goers.

The rest of the paper is organized as follows. The state of the art in terms of the impacts that AI can have on the transport sector is analysed, assuming a shift in the transport mode choice, within smart city networks and sustainable transport policies (Section 2). Based on such a review, a Smart Digital Solution Framework for urban mobility

is presented in Section 3. Then, Section 4 discusses an application case of an AI-based Mobility Control Centre for a medium-large city. Finally, some conceptual results and guidelines for government mobility levels are highlighted in Section 5. Conclusions and the road ahead are drawn and discussed in Section 6.

## 2. Impact of AI Paradigm Shifting on Transport

The concept of urban AI can be defined as follows: "Artifacts operating in cities, which are capable of acquiring and making sense of information on the surrounding urban environment, eventually using the acquired knowledge to act rationally according to pre-defined goals, in complex urban situations when some information might be missing or incomplete" [11].

The transport sector has entered an era of profound change, in which digital innovation, products, and services are substantially transforming the expectations, habits, and travel opportunities of people and freight. The sustainable and intelligent mobility market is rapidly evolving. Customers, operators, businesses, transport authorities, and governments are realizing and trying to exploit this huge potential. To seize all the opportunities and reduce the risk that innovation produces, inequalities and a modal shift that is not congruent with the objectives of sustainability and efficiency, cities need to aim for an integrated strategy with a system approach.

Therefore, with the aim to improve the users' travel experience as well as to promote a more sustainable and liveable city, the following sub-sections review the role of AI in supporting the promotion of dynamic and personalized mobility solutions and its contribution in planning sustainable mobility.

### 2.1. AI for Dynamic and Personalized Mobility

*The mobility demand is constantly evolving* [12]. The relationship between cities and mobility has assumed a central role in national and European strategies, where the objective is to face environmental challenges and raise the quality of life. Technologies enable cities to become smarter.

Mobility patterns have changed significantly in the last 10 years, and AI applied to new technologies will induce further significant changes. Innovations related to the collection and management of data and information (*IoT*—Internet of Things, mobile devices, wearable devices) and to the ever-increasing capacity and speed of analysis, simulation, and interpretation of phenomena will make the mobility system increasingly user-centric and adaptive.

This process is made possible by the use of Big Data Analytics, Machine/Deep Learning, and the automation of processes using AI (artificial intelligence) and IoE (Internet of Everything), making information in real time from multiple channels including mobile a reality. Also, widespread electronic payment systems (e-ticketing, EMV, Blockchain) have opened new opportunities in intermodal transport, which would not have been possible at such a scale in the recent past. Similarly, the freight movement will be based on an increasingly digital logistics chain and intelligent and sustainable urban distribution (city logistics, urban distribution centre, e-van sharing, delivery drones [2,13,14]).

It is a realistic forecast that the use of a private vehicle as the main transport mode in urban environments will give way to new intermodal mobility solutions: door to door, inclusive, shared, sustainable, and influenced by individuals. These are terms that will come to define the new mobility patterns. On the other hand, Generation *Y* is not less interested in car ownership; they are more interested in technology, as in the kind of devices they own [15,16]. Urban spaces are being reinterpreted and revised in the light of the changing needs of citizens.

*Several cities have set themselves ambitious sustainability goals, and the future has already arrived*. Some of the most advanced countries are already aiming, in their programs, at a drastic reduction in motorized vehicles, at the diffusion of charging stations for electric vehicles to encourage green modes of transport, at urban regeneration aimed at promoting

forms of sustainable mobility (pedestrianization, cycle paths, 30 Zone), and the construction of surface metro networks. There are around 675,000 bicycles and just 120,000 cars in Copenhagen, meaning bikes outnumber cars by more than five to one. Almost one-third of all journeys across the city are completed via bike, with Copenhageners cycling an estimated 1.44 million km (894,775 mi) every day.

Shared mobility solutions (such as car sharing and bike sharing) and on-demand transportation companies (such as Uber and Lyft) have already established themselves as reliable alternatives in contexts that envisaged demand management policies, such as the closure of historic centres and the disincentive for parking. Implementations of micro-mobility solutions are becoming popular as a way to bridge the "last mile" connectivity gap, especially for commuters [17–19].

From these contexts, the key concept behind the idea of "mobility-as-a-service" (MaaS) develops, [20]. MaaS has the ambition to put users, both passengers and freight, at the centre of transport services, offering them tailor-made mobility solutions based on individual needs. Thus, MaaS facilitates the integration of various modes of transport into a seamless travel sequence, with the possibility of bookings and payments (online and on the route) managed in a single transaction for all travel segments [21–25].

Furthermore, *the use of AI contributes to a faster implementation of solutions* for monitoring both the user choice behaviour and the performance of transport services. The use of AI also contributes to creating automation and decision support systems that enable *dynamic and personalized mobility management* [11,26–29]. Examples of AI methods that are successfully applied in the transport management field include the Fuzzy Logic Model (FLM) [28,30], Artificial Neural Networks (ANNs), Genetic Algorithms (GAs) [31], Simulated Annealing (SA), Artificial Immune System (AIS), the Ant Colony Optimizer (ACO), and Bee Colony Optimization (BCO) [26]. These AI algorithms have been experimented with in several applications for vehicles, infrastructure, drivers, or transport users, and, in particular, to enhance the dynamic interaction among each to deliver transport services that promote user empowerment, supporting human–machine interactions [26,32].

The benefits are undoubtedly for both the users and businesses. But the risks related to a undirected innovation tool that is too fast are just around the corner. The application of such innovations, many already underway, could lead to a new state of equilibrium in the urban mobility system, once again, far from the objectives of efficiency and sustainability. To give just one example, MaaS, without the governance of an appropriately calibrated smart direction maker, could fuel the disparity between users, i.e., those willing to pay more will be offered higher levels of service. And, moreover, in a possible scenario, it could lead to disincentivizing sustainable mobility itself. Services and apps for the rental of private vehicles and carpooling could encourage a shift towards the use of cars at the cost of local public transport. A further example is the implementation of a bike-sharing service, which is not accompanied by a connected and safe cycle network. The innovations require an intelligent government based on a dynamic system approach and, thus, planning based on data monitoring of demand.

*2.2. AI for Sustainable Urban Mobility Planning*

Planning plays a key role in the ability of cities to govern the innovation process of urban mobility without being subjected to it.

In order to reverse the trends and implement smart metropolitan cities, a significant challenge is to know how to implement a paradigm shift in the planning process.

The interpretation of the rapid changes in demand becomes crucial as well as the ongoing evaluation of construction choices. The implementation of a correct urban transport intervention, management, and planning policy has collided in the past with an often fragmented and incomplete knowledge of the mobility system in urban areas.

Cities, in the digital age, can adopt innovations that can revolutionize the way planning is carried out, overcoming the difficulties and limitations of the past. They can take advantage of the availability of big data and useful information for studying the behaviour

and habits of travellers in urban and metropolitan areas, for a realistic framework of the starting scenario as a basis for defining both the objectives to be pursued and the most suitable indicators that provide a quantitative verification of the effectiveness of the measures implemented to achieve the objectives themselves [11,26–28].

The adoption of the *Sustainable Urban Mobility Plan—SUMP* [33] is an opportunity for cities to start a profound reflection. In this context of difficult transition, a new perspective is needed to direct resources, energies, and intelligence. Integrated planning and a system approach take on a much-needed keystone for the promotion of sustainable mobility and efficient territorial governance.

"The SUMP is a strategic planning tool which, over a medium-long term time horizon (10 years), develops a vision of the urban mobility system (preferably referring to the metropolitan city area, where defined), proposing the achievement of environmental, social and economic sustainability objectives through the definition of actions aimed at improving the effectiveness and efficiency of the mobility system and its integration with the urban and territorial structure and developments". This is the SUMP definition of the ministerial decree, 4 August 2017, of the Ministry of Transport, which contains the guidelines for sustainable urban mobility plans, pursuant to art. 3, c.7 of Legislative Decree no. 257 of 16.12.2016 (*GU General Series n.233 of 05-10-2017*). This new approach to the strategic planning of urban mobility takes as its basis the document Guidelines Developing and Implementing a Sustainable Urban Mobility Plan (*ELTIS Guidelines*), approved in 2014 by the Directorate General for Mobility and Transport of the European Commission. The decree also establishes that "the definition of the objectives of the Plan and the monitoring of its implementation status must be based on solid quantitative evidence".

The SUMP not only constitutes one of the three indispensable administrative instruments for metropolitan cities to have access to state funding for the construction of new infrastructural interventions relating to mass rapid transport systems (metropolitan railway system, underground network) and a rewarding element in the context of other public investment programs (see, for example, the *Mobility as a Service for Italy Program* funded under the *Italian National Recovery and Resilience Plan—PNRR*; the *Multi-fund National Operational Program for Metropolitan Cities 2021-2027—PON METRO*) but it can also be an opportunity to initiate the paradigm shift necessary for strategic and integrated planning in cities [34].

To date, according to the SUMP Observatory, 206 cities in Italy have launched a planning process for sustainable mobility (70 approved; 53 adopted; 83 in progress). Among these, 10 are metropolitan cities: Milan, Turin, Bari, Reggio Calabria, and Messina have adopted the Plan; Bologna, Naples, and Genoa have approved it; Cagliari and Rome are in the design phase.

Cities that take the opportunity given by the adoption of *Sustainable Urban Mobility Plans* that will face problems in a new way and to equip the mobility systems with open data and service platforms will have models and tools available to systematically acquire information from the network and from the territory, and they will implement a governance dashboard of mobility phenomena and monitor the policies and choices implemented.

In accordance with the 11th Sustainable Development Goal (SDG) of Agenda 2030 (Sustainable Cities and Communities [35,36]), AI is not only useful in the transport operational phase but can support the development and monitoring of SUMP. The aim of this AI is to obtain a global vision of the city in terms of effective and sustainable mobility, understanding urban mobility phenomena, accessing data coming from scattered sources that are automatically collected and integrated, elaborating the big data to predict the impact resulting from applied or planned measures, and translate the impact of measures into mobility, health, and life quality indicators [11,26–28,37,38].

In the phase of the participatory process, in informing and listening to citizens' proposals for the design of the SUMP, AI applications support the necessary sharing of objectives and intervention scenarios for smart and community-friendly mobility.

To encourage this positive process, it is essential to make use of all possible innovations for the digital transformation of urban district mobility. A new type of mobility governance

supported by effective tools that can evolve over time and that are as open as possible in an increasingly digital, dynamic, interconnected, and intermodal ecosystem is a must, directing the creation of flexible, open, and user-centric solutions that start from data to create value and allow for the intelligent management of information to those responsible for the governance of mobility in urban and extra-urban contexts.

## 3. Smart Digital Solution Framework for Urban Mobility

The real difference comes from the evolution of the classic ITS in recent years into integrated digital platforms to support mobility from a smart city perspective [39–41]. In fact, in the Industry 4.0 era, implementing advanced technologies to enhance platform operations is crucial. According to [42], platforms can be basically categorized into (pure) product selling (PS) platforms and service platforms according to the functions they serve. Focusing on the latter, these digital platforms with advanced acquisition characteristics and great processing capacity of data from the territory and the transport system, rapid interpretation of phenomena and decision support for the management of the transport system in urban areas, orientation towards inter-modality, and of information multi-channels can add value to digitization and ITS investments and promote an effective and sustainable offer of E2E services.

To enable a digital ecosystem, the trend for modern metropolitan cities is to create AI-based Mobility Control Centres (MCCs), as a framework for data centralization and remote control of field systems for the management, regulation, and control of urban mobility, which integrates combinable systems and cooperative elements.

These elements and systems are (see Figure 1):

- *IoT*, e.g., traffic sensors, events' detectors; video cameras, Floating Car Data, telco data, drone data, all to recognize traffic and pedestrian flows, and critical events or disruption in urban areas for both linear infrastructures or terminal hubs (i.e., rail stations, ports, bus terminals, airports);
- *ITS and C-ITS*, such as traffic monitoring systems (TMSs); adaptive traffic control application; intelligent traffic light systems; system for monitoring travel time in an urban setting; restricted traffic areas and video surveillance systems, V2X solutions, and C-ITS; Smart Parking systems; advanced and multi-channel user information systems, etc.;
- *Big Data* solutions to acquire large amounts of heterogeneous data useful for historical and dynamic knowledge of the transport system and also for calibrating the models (evidence based);
- *Software for Traffic Simulation and Prediction* based on dynamic models for represent supply and demand system and their interaction, with methodologies and techniques of transport engineering;
- *AI* and algorithms based on transport models to interpret phenomena and support choices (DSS—Decision Support System) both in real time in the operational management of the mobility system and offline for transport service planning;
- *MaaS* must be integrated with the MCC to exploit their data and business logic (from a system optimum perspective) and convey to end users the dynamic offer of services, "best" for collective objectives, allowing for the choice of segments that make up the multi-intermodal journey, booking, payment, and use of customized advanced information related to the evolution of the journey and the surrounding real-time conditions (any disruption or changes to the network).

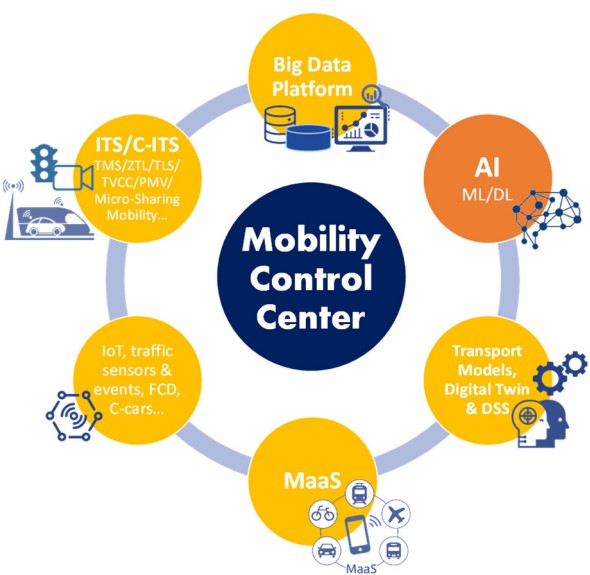

**Figure 1.** Main elements of Mobility Control Centre (MCC).

## 4. Results: Application Case in a Virtual District

The application case concerns the definition of a reference IT framework for a typical medium-large city that is already equipped with field sensors and ITS/C-ITS systems that has fragmented and non-dialoguing solutions and wants to acquire an AI-based Mobility Control Centre as a horizontal solution to integrate systems and services. This approach makes it possible to capitalize on previous investments and create value starting from the managed data. Such a process is ongoing and has been implemented in some medium-large cities in Italy (e.g., Reggio Calabria in Southern Italy). According to the 2023 Gartner Hype Cycle report, Digital Integration Hub (DIH) platforms are advanced application architectures that aggregate dispersed datasets in multiple backend record systems into a low-latency, high-performance, and scale-out data store, without unnecessary complexities. Starting from the DIH model introduced by Gartner in 2018 [43], this study proposes an architectural scheme for the Mobility Control Centre (MCC), as its specific declination within urban mobility with the AI add-on for monitoring and info-mobility.

The architectural scheme of the MCC includes (see Figure 2 from below to above/ bottom up):

1. *Data Source Integration Layer* that takes care of acquiring data from sensors and external field systems (even already existing ones) and all available sources through standard protocols and connectors realized ad hoc;

2. *High-Performance Data Layer* in which all phases of ingestion, pre-processing (data quality, anonymization, standardization, etc.), processing/elaboration, and storage of the data collected at the integration layer level take place. The data are organized in different databases according to whether they are hot data or cold data, used, respectively, for real-time or near-real-time processing during the operational phases and for offline processing to support planning;

3. *Business Logic Layer*, which is made up of the set of transport models and algorithms used to process the data, adding intelligence in the form of ML/AI/DL to create functions and services for the management of mobility. Typically, at this level, there is Software for Traffic Simulation and Prediction for the calculation of traffic flows in the network;

4. *API Management*, which consists of a scalable platform for creating, publishing, protecting, monitoring, and analysing APIs (application programming interfaces). The platform is intended to drive business goals, provide API-based services, and, at the same time, enable internal and external developers to adopt and integrate them easily

and quickly into their applications, facilitating interoperability with platforms and external systems;

5.  *Data Consumption Layer*, which represents the level of presentation and use of data and functions of the MCC for the end user (Institutions, Operators of the MCC, LPT Operators, maintenance workers, etc.), with profiled and customizable views. At this level, the Key Performance Indicators (KPIs) and Analytics Dashboards are also available to allow analysers, data scientists, and planners to investigate phenomena and critical elements in the mobility system.

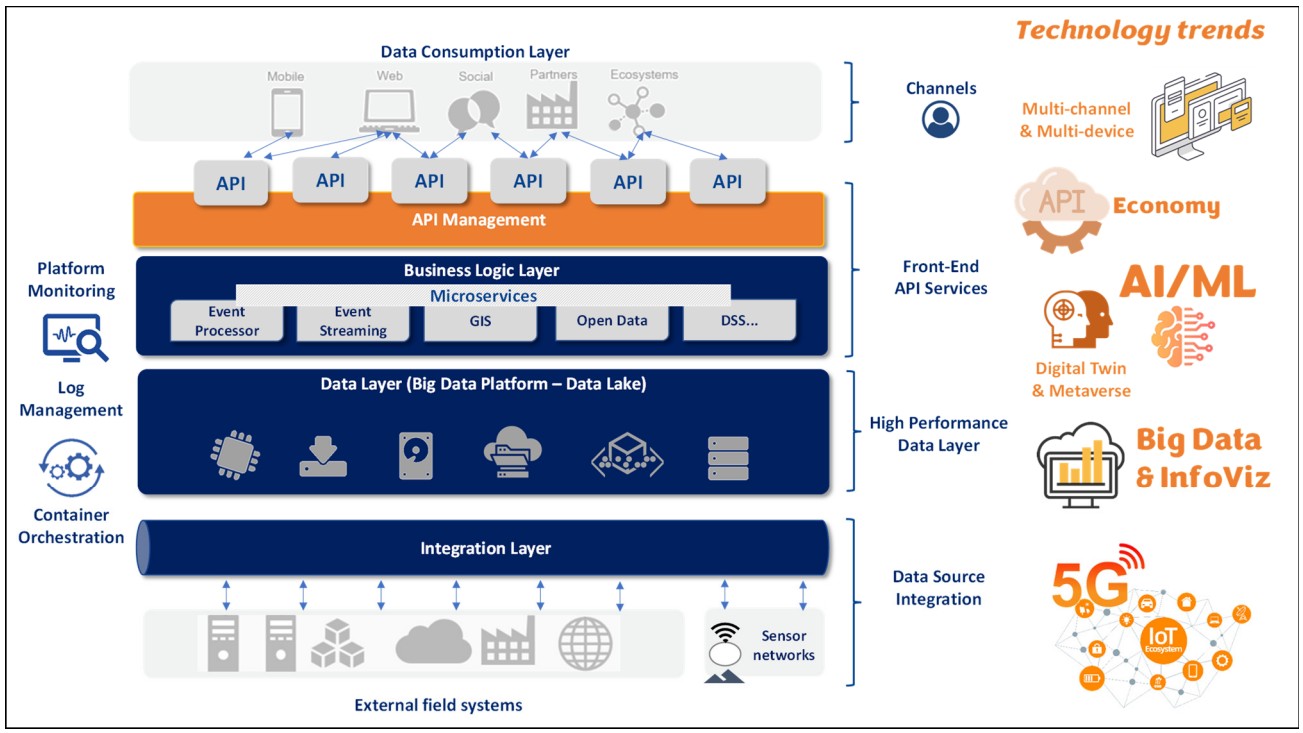

**Figure 2.** Architectural Scheme of Mobility Control Centre.

The operator interface (UI) of the MCC is composed of features and services organized into modules (macro-services) that must include, at least, the following (Figure 3):

*   *Supervision of traffic status* (A): This module must be equipped with a large set of tabular, graphical, and cartographic representations relating to the transport network (nodes, links, with a focus on the districts and routes of greatest interest) and supply system. The operator interface must allow for private traffic monitoring and the view, in a georeferenced way, of all the traffic events, the equipment, the peripheral devices, and the variables in the network's traffic conditions (congestion level, speed, travel time, and saturation), updated in real/near real time. The monitoring of local public transport, shared mobility (available/booked vehicles), and electric charging stations is also a feature of this module;

*   *Network and ITS Monitoring and Contro0l* (B): This module aims to provide an updated picture of the systems managed by the MCC—diagnostic and operational status—and allow remote control for the implementation of adaptive scenarios (usually integrated systems are limited traffic zone (LTZ) management systems, video surveillance systems, intelligent traffic light systems, VMS and information panels, smart parking systems, ramp metering, electric re-charging systems, etc.). The MCC UI must enable the operator to monitor traffic events from external systems (i.e., Waze) and to manually create, edit, and delete scheduled (construction sites, demonstrations) and unscheduled (i.e., accidents) events. For each ITS integrated and traffic event, general



information in accordance with the international and de facto standard (i.e., DATEX, GTFS, Transmodel, Netex, Siri, ETSI, etc.) must be managed;

- Decision Support System (DSS) (C) and what-if analysis: On the basis of the traffic forecasts defined using the STM models and on the basis of near-real-time data, the system is able to define the best distribution of the flows that engage the network. The outputs of the DSS are used to support the MCC operator in the management of the scenarios and remote actions to mitigate disruptions in the transport network (for example, opening/closing gates of the limited traffic zones (LTZs), modifying the traffic light plans, sending preconfigured messages on the VMS, activating video analysis systems) to manage the demand when events or disruptions occur in the network;
- Operational management and maintenance (D) through diagnostics and alarms; for each device displayed on the graphical interface, the diagnostic status (availability and alarms) of the individual components (e.g., LED of VMS, cameras, etc.), like, for each system, the operating status (e.g., message displayed, video streaming, etc.), is displayed;
- Prediction, KPIs, and analytics (E) through dashboards and information data visualization tools to allow analysts, data scientists, and planners to investigate phenomena and critical elements in the mobility system and understand the evolution and dynamics of the transport system.
- MCC asset management (F) through a device manager for remote management and updating of connected devices (sensors, info-mobility touchpoint, VMS) and the capability to define and control the layout and view of MCC assets (i.e., video wall, operator workstations, tactile tables, mixed-reality AR/VR viewers).

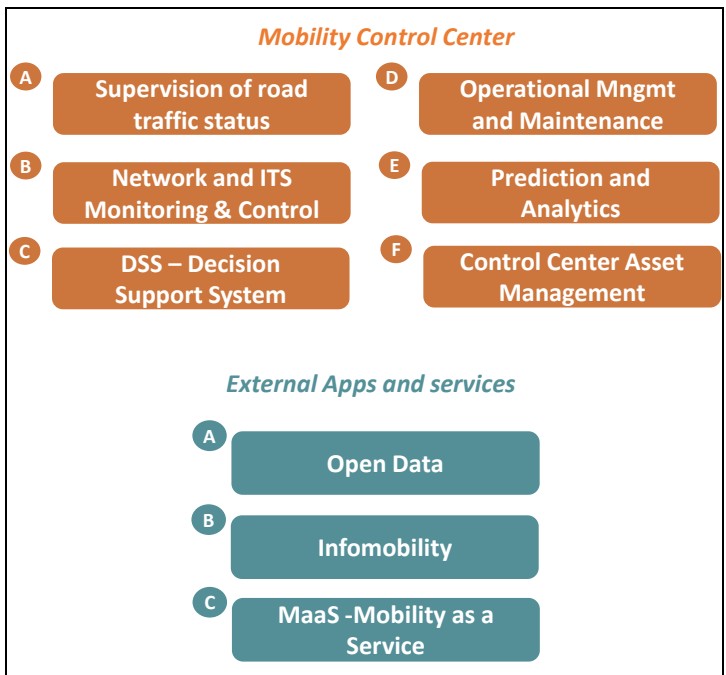

**Figure 3.** Mobility Control Centre macro-services and external apps.

For each module previously described, artificial intelligence makes it possible to automate and/or support various activities that MCC operators usually carry out manually or without the aid of intelligent information and suggestions. Based on an in-depth literature review, Table 1 summarizes the main module functions implemented in the developing MCC, and it highlights the AI/ML techniques that play and will play a crucial role to make these MCC solutions increasingly capable of making quick and agile decisions and automating the implementation of evidence-based mobility precision scenarios and policies.

**Table 1.** AI application for Mobility Control Centre.

| MCC Module/ Functionality | AI Application |
|---|---|
| *Supervision of road traffic status* | |
| *Traffic Flows Monitoring* | Video analysis solutions and AI algorithms for: (1) counting and tracking of pedestrian flows (counting people/objects crossing a virtual area, counting and density in a virtual area, anti-queuing and distancing, passenger flow, heat map); (2) road vehicles monitoring (in addition to the categorization of the vehicles and the colour, it is possible to detect: queue, reverse gear, U-turn, presence of vehicles in prohibited areas—e.g., pedestrian zones, reserved parking bays-, counting the number of vehicles cross a certain section) [44,45]. |
| *Private Traffic simulation and prediction* | Digital Twins (DT) combined with the availability of real-time traffic data and AI technology (focused on the end-to-end management and operation of global data) allow to predict traffic information [46,47]. DT provides the foundation for supplementing predominated (offline) microscopic simulation approaches with actual data to create a detailed dynamic digital representation of the physical traffic. DTs technology supports the construction of urban information and transport network models, including geographic information, new street view, real three-dimensional scene and ITS present on the transport network. Microscopic simulators of STM are used in modelling and simulating on-the-fly synchronized digital replicas of real traffic by leveraging fine-grained actual traffic data streams from road network traffic counters as input to the DT. The calibration features of microscopic simulators enable (dynamic) continuous calibration of running simulation scenarios. By doing so, the actual traffic data are directly fused into the running virtual model so that DT is continuously calibrated as the physical equivalent changes. Accordingly, DT and AI enables simulation-based control optimization during system run-time that was previously unattainable. Them, thus, forms the foundation for further evolution of real-time predictive analytics as support for safety–critical decisions in traffic management [46]. A recent study also explored traffic flow prediction methods using ML/DL techniques in autonomous vehicles and compared these models with respect to their applicability in modern smart transportation systems [47]. Finally, the integration of BIM (Building Information Modelling) and ITS, within smart transportation networks, facilitates operations such as the monitoring of intelligent road intersections and structural safety of bridges, in addition to the management of the maintenance activities [48]. The power of Artificial Intelligence and Machine Learning removes bottlenecks in the design process with BIM by automating repetitive tasks. |
| *Public Transport monitoring* | Recently it was explored the possibility of collecting anonymous data regarding public transport level of service, employing wireless technologies, big data and statistical filtering. AI/ML algorithms are used to detect, locate, create a map and record presence of travellers in public transport vehicles and stations, along with regular traffic data. The solution also serves as back-up system for locating vehicles on their path [49,50]. Another solution use data collected from AVL to predict future passenger demand on bus stops and routes using supervised machine learning techniques. The system can provide an accurate passenger demand prediction at bus stops [51]. |
| *Network and ITS Monitoring & Control* | |
| *Intelligent traffic lights systems* | Vision-based Traffic Signal Control: AI systems read stream live camera footage and adapt at an edge level the traffic signal plans to compensate, keeping the traffic flowing and reducing congestion. The system uses ML/DL reinforcement, where a program understands when it is not doing well (long queues at traffic lights) and tries a different course of action—or continues to improve when it makes progress [52]. Moreover, Genetic algorithm and fuzzy methods can be used to control the traffic signal systems automatically at intersections to adjust the waiting time, consequently the average waiting time for vehicles can be significantly reduced [53]. |
| *Limit Traffic Zone And Law Enforcement* | Limit Traffic Zones management based on vehicle detection and classification is one of the leading areas of research of Intelligent Transportation System. Among the technologies Artificial Intelligence (AI) has emerged as a giant in which vehicle classification has developed as a prominent subject of study because of its usefulness in several applications such as traffic control and surveillance, Law Enforcement in dense traffic environments, security systems and avoidance. Numerous algorithms and techniques for classifying vehicles have been proposed and implemented so far globally which mimics human intelligence [54]. Some solutions combine the best abilities of imaging, automated identification, radio communications and artificial intelligence technologies to identify not only vehicles with anomalous identities but also anomalous behaviour [55]. |

**Table 1.** *Cont.*

| MCC Module/ Functionality | AI Application |
|---|---|
| *Variable Message Signs Management* | Variable Message Signs (VMS) represent a cost-effective mechanism for disseminating information to drivers unequipped to receive personalized information. They can be used under incidents to divert traffic to less congested areas of the network to circumvent lengthy queues, better utilize network capacity, and improve system performance. Some VMS control algorithms seek diversion to enable a traffic system controller to favourably control traffic conditions in real-time, ensuring consistency with driver diversion response behaviour, being responsive to changing traffic conditions, enabling computational tractability through stage-based on-line implementation, and ensuring the spatial and temporal consistency of the displayed messages [56]. Other AI algorithms are used for the optimization models to find candidate roads for locating VMSs in real urban transportation networks [57]. A recent study proposed a prototype of a VMS reading system using machine learning techniques, for an ADAS (Advanced Driver Assistance System), which perceives the environment and provides assistance to the driver for his comfort or its safety. The assistant consists of two parts: a first that recognizes the sign on the road and another that extracts the text and transforms it into speech [58]. |
| *Smart Parking systems* | Smart parking system can use Image Processing and Artificial Intelligence: cameras and ultrasonic sensor deployed in locations to recognize the license plate numbers, ensuring ticketless parking. Big data analysis and neural network included in the algorithm provide related parking information and user recommendations [59]. AI can also help predict parking situations: for example, if there is a concert or other major event in town, AI can identify the areas that are most likely to be congested and recommend parking spots ahead of time. This would help drivers avoid traffic jams and save time. |
| *DSS—Decision Support System* | |
| *What If Analysis and Automated Scenarios* | AI technologies are used in ITS to monitor and manage transportation networks. This includes real-time incident detection, dynamic routing, adaptive traffic signal control, and providing traveller information to enhance safety and efficiency. AI can be used to automize the management of the scenarios and some remotely actions on ITS in case in which human decision is not required. For example, when low-impact scenarios have already been predefined which are activated when the pre-set thresholds of some variables are exceeded. In the event of queue phenomena on routes congested beyond a certain level, information is disseminated to impacted users to modify their driving behaviour, in order to mitigate disruptions on the transport network. Or to manage demand when scheduled events occur on the network such as marches or rallies. |
| *Demand-Responsive Transport* | Demand-responsive public transportation solutions also use artificial intelligence to optimize on-demand bus services which operate under flexible schedule and routes for both the driver and the passenger [60–62]. Many examples of these innovative services are operative in different metropolitan areas [63,64]. In particular, an advanced public transportation system called the local initiative for neighbourhood circulation (LINC) applies a genetic algorithm scheme to the optimization problem of the dial-a-ride service [65]. |
| *Operational management and maintenance* | |
| *Alarms Supervision* | AI and Big Data mixed technology provides various statistical indicators related to the type of alerts and occurrences registered by the MCC Modules through machine learning tools. It can automatically identify and prioritize alarms, while also displaying performance indicators related to the responsiveness in recognizing and resolving reported alarms. All indicators can be stored in the Data Layer and are later used by artificial intelligence algorithms. The data is correlated with contextual data (i.e., meteorological data) that may influence the occurrence of alerts. This functionality will allow the MCC Operator to optimize its response capacity since they will have a previous forecast on what issues may occur in their facilities in the future. |
| *Prediction, KPIs and Analytics* | |
| *Info Data Visualization Tool & Dashboarding* | The AI combined with Big Data tools and Info Data Visualization of the MCC supports the ex-post analysis of the key performance indicators about the service offered (load factor, travel times and commercial speeds, user satisfaction, energy consumption, sharing mobility performance, etc.) for the creation of dynamic thematic digital dashboards available to the various levels of mobility governance. Several studies analyse the capability of Data Visualization techniques using artificial intelligence for urban intelligent transportation scenarios com-paring the accuracy analysis of different visualization models [66,67]. |

**Table 1.** *Cont.*

| MCC Module/ Functionality | AI Application |
|---|---|
| *Mobile Edge Analytics* | The true potential of ITSs requires ultralow latency and reliable data analytics solutions that combine, in real time, a heterogeneous mix of data stemming from the ITS network and its environment. Such data analytics capabilities cannot be provided by conventional cloud-centric data processing techniques whose communication and computing latency can be high. Instead, edge-centric solutions that are tailored to the unique ITS environment must be developed. Recently an edge analytics architecture for ITSs was introduced in which data is processed at the vehicle or roadside smart sensor level to overcome the ITS's latency and reliability challenges. With a higher capability of passengers' mobile devices and intra-vehicle processors, such a distributed edge computing architecture leverages deep-learning techniques for reliable mobile sensing in ITSs. In this context, the deep-learning solutions and ITS mobile edge analytics capabilities pertaining to heterogeneous data, autonomous control, vehicular platoon control, and hyperphysical security were investigated [68]. |
| *MCC Asset Management* | In the MCC data can be collected from multiple sources ranging from roadside sensors, connected devices, embedded ITS, etc. These sources can be managed remotely through AI tools and asset management systems. These tool and systems, also using edge-based framework, allow massive configuration of sensors and devices, detect alarms or thresholds exceeded, and the support MCC operators in IoT-Based Predictive Maintenance, reducing resolution times of the disservices [69,70]. |

The MMC can also feed *open data* platforms by exposing the wide variety of information collected on the various elements of the mobility system to the outside, for example, data relating to the following: road vehicle flows, public transport performance and details of journeys made with respect to the scheduled plan, status and availability of mobility sharing and micro-mobility vehicles, scheduled and non-scheduled events, ITS operative status, etc.

By processing both traffic data and contextual events and integrating all the relevant information about the territory, the MCC can support, through artificial intelligence algorithms, intermodal, dynamic, and customized *Infomobility* for users [71].

## 5. Discussion

Some techniques from AI, information filtering agents (IFAs), and information monitoring agents (IMAs) can be used to evolve a population of personalized information. The technique of ML and the technique of learning from feedback can be combined to develop a semi-automated information filtering system, which dynamically adapts to the changing interests of the user, maximizing satisfaction during the journey. Furthermore, this could support demand management policies in a system optimum logic approach, both in ordinary and emergency network conditions.

Moreover, we must highlight the potential of new incoming AI tools, such as ChatGPT [72]. ChatGPT facilitates customer interactions and provides personalized and dynamic recommendations. ChatGPT, developed by OpenAI, is a natural language processing (NLP) system that can understand and generate text-based conversations, making the way transportation companies interact with customers easier, and they can also access data more easily. ChatGPT can suggest the most efficient and environmentally friendly paths and offer advice on public transportation and other forms of mobility, including micro-mobility, cycling, and walking. ChatGPT can also be used to promote public education and awareness about sustainable transport and mobility. It can provide users with detailed information about the benefits of sustainable transport and mobility, as well as tips and tricks for reducing their carbon footprint [73].

MaaS (Mobility as a Service), a new transport concept with limited applications and only partial implementation so far, utilizes personalized mobility, which has the potential to replace privately owned vehicles and optimize the use and combination of several mobility alternatives. MaaS can maximize its potential if it is based on an IT framework like the Mobility Control Centre that incorporates all data sources and relevant information from a

smart urban district to propose mobility alternatives to users that optimize the transport network in holistic way.

MaaS should be enabled by powerful AI algorithms, which provide holistic travel planning, booking and ticketing, and real-time information services, customized and tailored to each consumer's needs. For this purpose, as previously mentioned, ChatGPT technology can be used to answer customer questions, such as those regarding the availability of transportation services, the best routes, less congested parking areas, and the cost of travel, and guide them towards more sustainable transport and intermodal mobility options. It is designed to provide tailored advice to users, based on their preferences and current context.

Last but not least, artificial intelligence, Machine Learning, and Augmented Reality (AR) technology are already enhancing the operations of *autonomous and semi-autonomous vehicles*, as well as other vehicles equipped with advanced driver assistance systems (ADASs) that help drivers to drive, reverse, and park safely [47]. This next generation of AI and AR advancements has the power to push driver assistance technology even further and give consumers a more predictive and personalized driving experience. The MCC, onboard AI/ML, and AR can cooperate to achieve optimal driver personalization and anomaly prediction. Both supervised and unsupervised learning must be employed to create a system that can observe and learn equally from behaviour behind the wheel, vehicle sensors, and the urban transport network. AI-powered vehicle systems capture the gestures and emotions of drivers and passengers to predict behaviour and implement incremental learning. They also predict contextual events and issues by fusing multi-modal inputs derived from cameras, image frames, and sensory data to reach a specific conclusion. The cooperative system generates personalized driving recommendations based both on the specific habits of drivers and the suggested actions by the MCC based on the real-time traffic conditions. Finally, AI-based crash detection mobile systems (on smartphones and wearable devices) can be used to automatically send information to emergency services after a severe car crash and can also enable automatic voice connection if injured passengers are unresponsive [74]. These data, if transferred to the MCC, inform and alert the control centre, allowing one to track and manage the event, including assistance.

## 6. Conclusions

In this paper, a Mobility Control Centre (MCC) and artificial intelligence (AI) framework for urban sustainable districts are presented. Urban AI is part of a greater stream of the digital transformation of the intertwined physical, social, and digital realities. On the operational side, the sharing of infrastructure (sensors, hardware, software) and data is key for urban AI business cases.

AI enables applications to aid intelligent transportation [6]; however, it faces limitations. There is an important role in the human factor to maximise the benefit that can be obtained in applications of AI solutions and avoid undesirable outcomes. AI cannot become smarter than its training datasets; therefore, the issues related to the collection and storage of standardised data become crucial. On the other hand, overly strong data protection regulations or inadequate frameworks on data ownership and usage may slow down the progress of AI technologies. For example, all public transport stakeholders have a role to play including end users because AI is a living application, which learns from user input, with the potential to take over daily and repetitive tasks. All the above could be barriers if they are not well pointed out and could limit the vision of stakeholders, who should proactively look at the opportunities brought by AI to improve their services and build the mobility of tomorrow. Therefore, according to [75], although there are many advantages in using AI for improving mobility, the limitations in the subfields of AI need to be considered. Similar advantages across all subfields include increased cost and time savings, improved safety, better accuracy, and overall increased productivity. Some of the limitations of AI subfields include incomplete data, high initial cost of deployment,

data and knowledge acquisition issues, as well as consolidated methodologies for proving real-time advice to users for travelling as well as to operators for operational control [76].

In "Big data", recording and processing and bi-directional communication between travellers and MCCs are emerging as the two factors that can improve the performances of systems that support short-term forecasting of the network status. Nevertheless, the methodologies applied in current tools are not fully up to the level of the research in the field of transit network modelling, and further effort is needed in the application and development of real-time on-board load short-term forecasting, real-time best path suggestions, real-time transit assignment modelling, individual path choice modelling, and real-time reverse assignment. The application of AI solutions has met obstacles and difficulties, due to a lack of detail in the regulations on data use, especially as most of the existing regulations were written not considering learning processes. Future intelligent systems will provide real-time data from passengers' and cars' sensors to the transportation sector. Overall, artificial intelligence can assure better travel experiences.

AI also offers solutions for anticipating weather and traffic patterns, managing roads, and alerting on-duty police officers. Before starting their trip, these systems support drivers, commuters, and pedestrians. It is important to have technological support in order to create an effective public transportation system that aids in planning and decision making.

Every change, even the most positive one, as is the case of smart cities and urban mobility, implies costs. Those costs are not necessarily evenly and fairly distributed among those who benefit from the changes; therefore, there are many challenges to be addressed and resolved.

Only a comprehensive and digital approach to urban transport can be effective in translating current mobility management into sustainable mobility management for cities, reducing the risk of fragmented policies, which lead to non-resolutions.

Local public administrations should, therefore, develop and adopt SUMPs and, in accordance with the planned objectives, invest in the creation of AI-based MCCa, therefore enabling technological frameworks and operational personnel with domain and transversal skills. AI-based Mobility Control Centre services should be designed and implemented to simplify and optimize both the planning phase of transport services and their dynamic management, also taking the opportunity to review non-performing and uneconomical organizational processes. These services should be implemented without ever neglecting regulatory and safety indications (i.e., the European regulation on artificial intelligence—AI Act, approved by Parliament and which will enter into force in 2024) to guarantee an even more equitable and inclusive mobility system.

The challenge of the digital transition for urban transport and the application of artificial intelligence can enable a paradigm shift necessary for enhanced quality of life in our cities. However, a clear planning framework, a Mobility Control Centre integrated with AI-based services and MaaS platforms, and specialized human capital for effective and *precision government* are needed.

**Author Contributions:** Conceptualization, F.M.M.C., A.C. and A.Q.; methodology, F.M.M.C., A.Q. and A.C.; software, F.M.M.C. and A.Q.; writing—original draft preparation, F.M.M.C., A.Q. and A.C.; writing—review and editing, F.M.M.C., A.C. and A.Q.; visualization, F.M.M.C., A.Q. and A.C.; supervision, F.M.M.C., A.C. and A.Q. All authors have read and agreed to the published version of the manuscript.

**Funding:** This research received no external funding.

**Data Availability Statement:** Not available.

**Acknowledgments:** The authors wish to thank the reviewers for their contribution, which were highly useful in improving the paper, and MDPI for the APC waiver.

**Conflicts of Interest:** The authors declare no conflict of interest.

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
