# Peer review of "Mobility Control Centre and Artificial Intelligence for Sustainable Urban Districts"

_information, doi:10.3390/info14100581_

Round 1

Reviewer 1 Report

Dear authors,

concepts of mobility control center based on AI it has been studied by other researchers too. Therefore, my biggest concern is related to the novelty of your research. What gap in the field does your research precisely bridge? Can you be more concrete about why your research is important compared to other existing studies? It seems that you just provide a literature review, highlighting that AI is a magic stick that will solve all mobility problems. Personally from my research perspective that is not always the case in mobility, AI has prone and cone in mobility applications.

Therefore, the manuscript requires significant improvements, I recommend next: 

1. Improve Abstract (it should reveal what did you find that other studies not include, or what did you do differently from existing studies) 

2. Improve Introduction should summarise the motivation for your research, what is done, and what are significants of your findings in the field.

3. Give a Literature review chapter that reveals all relevant studies (state of the art in the field) in the field, based on that state why, which gap you will bridge

Further, 

you presented the concept of MCC and its architectural scheme. Did you add some novelty in the logic of the concept, layers compared to other studies? This should be highlighted if soo and explained in depth why you think is important. 

Best

Some abbreviations seem to be not defined in the text, and some words are improperly written so check all text carefully once again.

Author Response

Dear authors,

concepts of mobility control center based on AI it has been studied by other researchers too. Therefore, my biggest concern is related to the novelty of your research. What gap in the field does your research precisely bridge? Can you be more concrete about why your research is important compared to other existing studies? It seems that you just provide a literature review, highlighting that AI is a magic stick that will solve all mobility problems. Personally from my research perspective that is not always the case in mobility, AI has prone and cone in mobility applications.

ANSWER

Thank you very much for raising concerns. We are grateful for your valuable feedback. We agree with your comment. Your comment has been taken into careful consideration, We have taken your advice  to stress the innovation  to propose to the current state of the art, and how it is guiding the development of a mobility control centre in a medium-size city. We revised the paper to incorporate the considerations you suggested.

“…

Starting from an in-depth and model-based analysis of the application cases for the different automation techniques, the main objectives of this  paper are to review the current state of the practice of AI and telematics in supporting the improvement of the sustainability and livability of urban and metropolitan areas, to explore the capacity of AI in building smart digital solutions, also elaborating some key recommendations for city scientists, policymakers, transport and urban planners and an agenda for future research directions, as well as to present how such findings are guiding the development of an AI-based Mobility Control Center in a medium-large city.

Therefore, the results will show how AI is bringing the urban mobility sector closer through Mobility-as-a-Service (MaaS) schemes and illustrate with a virtual district how such AI-enabled technological advances linked with public policies can help design sustainable urban mobility for providing a better travel experience to city users.

…”

Therefore, the manuscript requires significant improvements, I recommend next:

  1. Improve Abstract (it should reveal what did you find that other studies not include, or what did you do differently from existing studies)

ANSWER

We are very thankful and we deeply valued the comprehensive comments The abstract was revised as suggested, pointing out the innovation provided by such a paper.

 “…

The application of Artificial Intelligence (AI) to mobility dynamic management can support the achievement of efficiency and sustainability goals. AI, by processing big data from heterogeneous sources in a very short time, can help in the real-time diagnosis of the mobility system, and by comparing phenomena in similar contexts, identify network and service configurations, as well as support the implementation of measures for managing demand that allows achieving the sustainable goals. It this paper, an in-depth analysis of scenarios, IT (Information Technology) framework based on emerging technologies and AI to support sustainable and cooperative digital mobility is provided. Therefore, the definition of the functional architecture of an AI-based mobility control center is defined and the process that has been implementing in a medium-large city is presented.

…”

  1. Improve Introduction should summarise the motivation for your research, what is done, and what are significant of your findings in the field.

ANSWER

Thank you for your comments and valuable feedback. we have revised the Introduction accordingly, by incorporating additional pertinent statements, we have strived to enhance the comprehensiveness of the Introduction.

  1. Give a Literature review chapter that reveals all relevant studies (state of the art in the field) in the field, based on that state why, which gap you will bridge

ANSWER

We are grateful for your valuable feedback. We agree with your comment. We have reviewed state of the art studies, to enhance clarity and strengthen the scientific contributions of the paper.

 “…

Therefore, with the aim to improve the city users’ travel experience as well as to promote a more sustainable and liveable city, the following sub-sections review the role of AI in supporting the promotion of dynamic and personalized mobility solutions, and its contribution in planning the sustainable mobility.

…”

Further,

you presented the concept of MCC and its architectural scheme. Did you add some novelty in the logic of the concept, layers compared to other studies? This should be highlighted if soo and explained in depth why you think is important.

ANSWER

Thank you for your comment. We have revised and added as suggested

Reviewer 2 Report

As urban populations continue to grow, city planners and administrators are increasingly leveraging AI to help create more efficient, sustainable and livable urban environments. In this context, artificial intelligence infrastructure becomes a key element of the future of smart cities. One of the most important ways that AI is transforming urban planning is through the use of data. The reviewed article is devoted to this issue. I have some questions and comments about the manuscript. Were the examples of using AI in the Mobility Control Center presented by the authors selected on the basis of research and analysis? Or maybe the presented application example should be treated rather as a review of available technologies used by smart cities? If so, please indicate in the abstract that this is a review of available technologies.

Author Response

As urban populations continue to grow, city planners and administrators are increasingly leveraging AI to help create more efficient, sustainable and livable urban environments. In this context, artificial intelligence infrastructure becomes a key element of the future of smart cities. One of the most important ways that AI is transforming urban planning is through the use of data. The reviewed article is devoted to this issue. I have some questions and comments about the manuscript. Were the examples of using AI in the Mobility Control Center presented by the authors selected on the basis of research and analysis? Or maybe the presented application example should be treated rather as a review of available technologies used by smart cities? If so, please indicate in the abstract that this is a review of available technologies.

ANSWER

Thank so much for your comments and for having pushed us to stress to what the paper is devoted. The abstract, introduction and the Section 2 (state of the art) have been revised in such a direction.

Reviewer 3 Report

The paper is a form of discussion on mobility in the face of technological changes, or rather - what modern mobility must face in the context of innovations that surround us.
However, this is just a discussion of the literature. There are few specifics here that would be interesting, nothing new. What does the author really offer the reader?
From the editing side, I don't understand the use of bold text in several cases. The language should be a bit more formal - it's not a press release in a newspaper, it's a science paper and a journal like that.
In my opinion, the Author(s) should think about what they want to say, measure the phenomenon, show the tested solution, prepare discussion.

Author Response

The paper is a form of discussion on mobility in the face of technological changes, or rather - what modern mobility must face in the context of innovations that surround us.

However, this is just a discussion of the literature. There are few specifics here that would be interesting, nothing new. What does the author really offer the reader?

ANSWER

Thank so much for your comments and for having pushed us to stress the innovation that we would like to propose to the current state of the art, and how it is guiding the development of a mobility control centre in a medium-size city. We revised the paper to improve as suggested.

From the editing side, I don't understand the use of bold text in several cases. The language should be a bit more formal - it's not a press release in a newspaper, it's a science paper and a journal like that.

ANSWER

We appreciate the valuable suggestions provided. In response to the raised points, we have diligently carried out modifications in the revised manuscript, incorporating the suggested changes to enhance clarity and strengthen the scientific contributions of the paper.

In my opinion, the Author(s) should think about what they want to say, measure the phenomenon, show the tested solution, prepare discussion.

ANSWER

Thank you very much for raising concerns. We have revised the paper accordingly

Reviewer 4 Report

The manuscript titled as "Mobility Control Center and Artificial Intelligence for Urban Sustainable Districts" is interesting, but research rigour and motivation are quite unclear. Authors should strengthen the scientific value of this study, and refer to my following comments:

- In Abstract, the findings and insights from this work should be summarised and presented. 

- What are the research problems/gaps from this study? 

- The research questions should be clearly listed in Section 1. 

- In Section 3, authors should refer to similar existing studies to construct your own solution framework, for example:

Towards generic platform to support collaboration in freight transportation: taxonomic literature and design based on Zachman framework. Enterprise Information Systems17(2), 1939894. (2023). 

Artificial intelligence in industrial design: A semi-automated literature survey. Engineering Applications of Artificial Intelligence112, 104884. (2022). 

The core elements in your proposed framework are quite unclear. To me, authors merely put all emerging technologies together without detailed explanation. Authors should revise the framework with showing its value and motivation. 

- The background and motivation of your application case should be further elaborated. 

- Your descriptions in Table are like the summarisation and literature review. If this work is a review work, authors should re-organise the whole work. 

- What is the novelty of this work?  

Authors should proofread and edit the manuscript before re-submission. A number of inconsistency and grammatical mistakes are found in this work. If possible, this manuscript should be sent to editing by Native English editors. 

Author Response

The manuscript titled as "Mobility Control Center and Artificial Intelligence for Urban Sustainable Districts" is interesting, but research rigour and motivation are quite unclear. Authors should strengthen the scientific value of this study, and refer to my following comments:

- In Abstract, the findings and insights from this work should be summarised and presented.

ANSWER

Thank you very much for your comment. The abstract was revised as suggested, pointing out the innovation provided by such a paper.

“…

The application of Artificial Intelligence (AI) to mobility dynamic management can support the achievement of efficiency and sustainability goals. AI, by processing big data from heterogeneous sources in a very short time, can help in the real-time diagnosis of the mobility system, and by comparing phenomena in similar contexts, identify network and service configurations, as well as support the implementation of measures for managing demand that allows achieving the sustainable goals. It this paper, an in-depth analysis of scenarios, IT (Information Technology) framework based on emerging technologies and AI to support sustainable and cooperative digital mobility is provided. Therefore, the definition of the functional architecture of an AI-based mobility control center is defined and the process that has been implementing in a medium-large city is presented.

…”

- What are the research problems/gaps from this study?

- The research questions should be clearly listed in Section 1.

ANSWER

We appreciate the valuable suggestions provided. We have revised as suggested.

“…

In this context, the AI-based solutions can push towards a significative advancement to city mobility in all cities within all countries. They give win-win results when the same device systems are used in a cooperative glance among different stakeholders. Some base features, such as collecting data, storing data, and analyzing data can be integrated in order to improve the existing urban transport systems. Subsequently, the open questions to which the paper wants to contribute are the following:

  • How can the AI contribute to promote the urban sustainable mobility improving the users’ travel experience and shifting towards more sustainable transport means? (Section 2)
  • how can the urban sustainable mobility planning benefit from the opportunity offered by AI in supporting decision making for identifying and implementing sustainable measures? (Section 3)
  • which are the main features of a mobility control center able to promote sustainable mobility? (Sections 4 and 5)

Therefore, starting from an in-depth and model-based analysis of the application cases for the different automation techniques, the paper reviews the current state of the practice of AI and telematics in supporting the improvement of the sustainability and livability of urban and metropolitan areas, explores the capacity of AI in building smart digital solutions, also elaborating some key recommendations for city scientists, policy-makers, transport and urban planners and an agenda for future research directions, as well as presents how such findings are guiding the development of an AI-based Mobility Control Center in a medium-large city.

The results will show how AI is bringing the urban mobility sector closer through Mobility-as-a-Service (MaaS) schemes and illustrate with a virtual district how such AI-enabled technological advances linked with public policies can help design sustainable urban mobility for providing a better travel experience to city users.

The rest of the paper is organized as follows. The state-of-the-art on the impacts which AI can have on the transport sector is analyzed, in assumption of a shift of transport mode choice, within smart cities networks and sustainable transport policies (Section 2). Based on such a review, a Smart Digital Solution Framework for urban mobility is presented in Section 3. Then, Section 4 discusses an application case of an AI-based Mobility Control Center for a medium-large city. Finally, some conceptual results and guidelines for mobility government levels are highlighted in Section 5. The conclusions and the road ahead are drawn in Section 6.

…”

- In Section 3, authors should refer to similar existing studies to construct your own solution framework, for example:

Towards generic platform to support collaboration in freight transportation: taxonomic literature and design based on Zachman framework. Enterprise Information Systems, 17(2), 1939894. (2023).

Artificial intelligence in industrial design: A semi-automated literature survey. Engineering Applications of Artificial Intelligence, 112, 104884. (2022).

ANSWER

We are grateful for your valuable feedback. We have revised as suggested.

The core elements in your proposed framework are quite unclear. To me, authors merely put all emerging technologies together without detailed explanation. Authors should revise the framework with showing its value and motivation.

ANSWER

Thank you for your comments. We have revised the paper to enhance the comprehensiveness of the conclusions

- The background and motivation of your application case should be further elaborated.

ANSWER

Your comment has been taken into careful consideration, and we have revised the application section.

- Your descriptions in Table are like the summarisation and literature review. If this work is a review work, authors should re-organise the whole work.

ANSWER

Thank you for your comments. We have revised the paper to enhance the comprehensiveness of the conclusions

- What is the novelty of this work?

ANSWER

Thank you very much for raising concerns. The Introduction has been revised in order to point out the novelty of this research.

Round 2

Reviewer 1 Report

Dear authors, you made improvements to the manuscript. The research topic is interesting and relatively contributes to the field. However, I'm still not sure about the strength of the novelty your paper delivers in comparison to similar works. My score: Overall manuscript rating (weak accept)

My advice for authors is not to just elaborate on the possible potential of AI in decision-making in mobility but also to address the actual challenges that limit some subfields of AI application in a real system (in real-time). These challenges actually exist and must be taken into account when talking about AI-based decision-making systems in Mobility Control Centre.

Minor editing of English language required

Author Response

REVIEWER 

Dear authors, you made improvements to the manuscript. The research topic is interesting and relatively contributes to the field. However, I'm still not sure about the strength of the novelty your paper delivers in comparison to similar works. My score: Overall manuscript rating (weak accept)

My advice for authors is not to just elaborate on the possible potential of AI in decision-making in mobility but also to address the actual challenges that limit some subfields of AI application in a real system (in real-time). These challenges actually exist and must be taken into account when talking about AI-based decision-making systems in Mobility Control Centre.

ANSWER

Thank you so much for your valuable comments. We have reviewed and highlighted out criticalities that need to be considered when implementing AI applications.

Reviewer 2 Report

The authors made changes to the article.

Author Response

Thank you for your comments, which we found very useful in reviewing our paper.

Reviewer 3 Report

In my opinion, there is a lack of solid discussion related to the topics described in chapters 2 and 3. At the end we see a positive summary of these parts, but in my opinion critical look and at least a short discussion about the challenges of AI face in this particular application are needed.
I also miss the same thing in the summary. Here, the usual, simplest SWOT analysis is desirable.

Author Response

REVIEWER 

In my opinion, there is a lack of solid discussion related to the topics described in chapters 2 and 3. In the end, we see a positive summary of these parts. Still, in my opinion, a critical look and at least a short discussion about the challenges AI faces in this particular application are needed.

I also miss the same thing in the summary. Here, the usual, simplest SWOT analysis is desirable.

ANSWER

Thank you for your comment. We have taken the advice and in reviewing have added a critical overview in order to capture the challenges and the limitations of AI.

Reviewer 4 Report

The manuscript, titled as "Mobility Control Center and Artificial Intelligence for urban sustainable districts", proposes a theoretical framework about the city mobility. In my opinion, the framework establishment lacks the theoretical support. Authors might consider formulateing the research framework based on the extant literature. 

- The motivation of needing this framework is unclear. What is the problem now when we do not have the framework? 

- Authors should refer to some systematic review methods to formulate the research framework, e.g. 

Tsang, Y. P., & Lee, C. K. M. (2022). Artificial intelligence in industrial design: A semi-automated literature survey. Engineering Applications of Artificial Intelligence112, 104884.

Siqin, T., Choi, T. M., Chung, S. H., & Wen, X. (2022). Platform operations in the industry 4.0 era: recent advances and the 3As framework. IEEE Transactions on Engineering Management.

- What are the emerging research issues to be addressed in future? Any future research agenda based on this framework? 

- Authors should also consider the government policies to enrich your framework. I am questionnable about the practicality or theoretical contribution of the proposed framework. 

The language used in this manuscript should be fine. 

Author Response

REVIEWER 

The manuscript, titled as "Mobility Control Center and Artificial Intelligence for urban sustainable districts", proposes a theoretical framework about the city mobility. In my opinion, the framework establishment lacks the theoretical support. Authors might consider formulating the research framework based on the extant literature.

ANSWER

Thank you for your comments. We have gone further in depth on the framework analysis, in doing so your suggested paper have been very useful in formulating the research network.

- The motivation of needing this framework is unclear. What is the problem now when we do not have the framework?

ANSWER

Thank you, as in the previous answer, we reviewed the framework as suggested.

- Authors should refer to some systematic review methods to formulate the research framework, e.g.

Tsang, Y. P., & Lee, C. K. M. (2022). Artificial intelligence in industrial design: A semi-automated literature survey. Engineering Applications of Artificial Intelligence, 112, 104884. GIA’ CE LO AVEVA SEGNALATO!!!

Siqin, T., Choi, T. M., Chung, S. H., & Wen, X. (2022). Platform operations in the industry 4.0 era: recent advances and the 3As framework. IEEE Transactions on Engineering Management.

ANSWER

Done

- What are the emerging research issues to be addressed in future? Any future research agenda based on this framework?

ANSWER

Thank you, we have added some considerations in regards in the Conclusion paragraph.

- Authors should also consider the government policies to enrich your framework. I am questionnable about the practicality or theoretical contribution of the proposed framework.

ANSWER

Thank you for the comment, we have included policies in the framework, and considerations on theoretical contributions.

Round 3

Reviewer 1 Report

Dear authors, thanks for considering my suggestions, the paper seems better now.

In line 521 remove the letter g in front of the word better "gbetter travel experiences"

Thanks

Author Response

Thank you for your comments and suggestions. We have corrected line 512.

Reviewer 4 Report

After several rounds of revision, the manuscript is now up to the acceptance bar. I have no further comment about this study. Good luck. 

The use of English is fine in this work. 

Author Response

Thank you for your comments and valuable feedback. It contributed to improve the output of our paper.